# Using BIBTEX to Automatically Generate Labeled Data for Citation Field Extraction

**Dung Thai**                                            DTHAI@CS.UMASS.EDU
**Zhiyang Xu**                                     ZHIYANGXU@CS.UMASS.EDU
**Nicholas Monath**                                   NMONATH@CS.UMASS.EDU
*University of Massachusetts Amherst*

**Boris Veytsman**                      BVEYTSMAN@CHANZUCKERBERG.COM
*Chan Zuckerberg Initiative*

**Andrew McCallum**                                   MCCALLUM@CS.UMASS.EDU
*University of Massachusetts Amherst*

## Abstract

Accurate parsing of citation reference strings is crucial to automatically construct scholarly databases such as Google Scholar or Semantic Scholar. Citation field extraction (CFE) is precisely this task—given a reference label which tokens refer to the authors, venue, title, editor, journal, pages, etc. Most methods for CFE are supervised and rely on training from labeled datasets that are quite small compared to the great variety of reference formats. BIBTEX, the widely used reference management tool, provides a natural method to automatically generate and label training data for CFE. In this paper, we describe a technique for using BIBTEX to generate, automatically, a large-scale (**41M** labeled strings), labeled dataset, that is four orders of magnitude larger than the current largest CFE dataset, namely the UMass Citation Field Extraction dataset [Anzaroot and McCallum, 2013]. We experimentally demonstrate how our dataset can be used to improve the performance of the UMass CFE using a RoBERTa-based [Liu et al., 2019] model. In comparison to previous SoTA, we achieve a **24.48%** relative error reduction, achieving span level F1-scores of **96.3%**.

## 1. Introduction

Scholarly knowledge bases, such as Google Scholar and Semantic Scholar, are invaluable tools for navigating the landscape of scientific literature. Scholarly knowledge bases critically depend on the accurate bibliographic information to both populate research paper records (containing information such as author, publication venue, publisher) and provide citation graph information of which papers cite which other papers. While some such information is manually curated in sources such as PubMed, Web-of-Science, and others[1][2], the services are often incomplete (missing tech reports, preprints, some books and conference proceedings) or are not publicly available. Other services (such as Google Scholar, Semantic Scholar, and CiteSeer) aim to cover a wider set of academic papers by extracting relevant bibliographic information from a massive amount of unstructured sources collected from a variety of sources (e.g., web-crawls, conference proceedings). The quality of the information extraction tools impacts the quality of the database and in turn the user experience of the product and, as a result, the user's research efficiency and effectiveness.

---

1. https://www.crossref.org/
2. https://i4oc.org/

A core component of the automatic extraction of bibliographic information is *citation field extraction*, the task of segmenting a citation or reference string [3] into constituent parts (fields), such as title, author(s), publisher, and year. This also includes relatively uncommon fields such as language indicator, document type, and organization names. The size of the training data in the benchmark citation field extraction datasets is relatively small. The two mostly widely used labeled benchmarks, the most widely used labeled benchmarks, UMass Citation Field Extraction dataset [Anzaroot and McCallum, 2013] contains 2476 references. However, references appear in a multitude of different formats throughout sciences and are often manually entered by scientists increasing the variability of their structure and formatting.

Data augmentation [Tran et al., 2017, Ratner et al., 2017] and other automatic training data generation methods [Tripathi et al., 2019] have been shown to be highly effective in settings with limited training data. Examples of this include: closed captioned videos have been used to create automatically labeled speech recognition data [Lakomkin et al., 2018], the automatic generation of 3D point cloud shapes has been used for self-driving vehicle systems [Yue et al., 2018], and program synthesis to improve information extraction from political data and flight emails [Iyer et al., 2019]. Citation field extraction is uniquely well suited for these techniques using reference formatting tools, namely BIBTEX. BIBTEX can be used to format references in a wide variety of formatting styles and fonts, while using the structure of BIBTEX records to automatically label the reference fields for training and evaluating citation field extraction models. This provides us a mechanism to automatically generate labeled data.

In this paper, we build a large-scale citation field extraction dataset which is automatically generated from publicly available BIBTEX files. We collect human-curated bibliographies from writing project repositories, authors and websites are collected and randomly paired with various styles to produce a set of reference strings. The references are aligned with their labels provided in the BIBTEX files to form labeling sequences. We collected 41M labeled references in total, twenty thousand times the size of the UMass CFE dataset.

We train a variety of citation field extraction models including one based on RoBERTa [Liu et al., 2019]. We show that this model trained only on the UMass CFE dataset matches state-of-the-art results [Thai et al., 2018]. We then show that training the BERT-based model on our large automatically generated dataset drastically improves the results, outperforming the state of the art approach by 1.2 points of F1, a 24.48% relative reduction in error. We then: show that certain subsets of our automatically generated dataset are considerably more challenging than existing benchmarks, present these datasets for evaluation in future work, and analyze our experimental results. All data and models for our approach are publicly available. [4]

## 2. Citation Field Extraction

Citation field extraction (CFE) is the task of segmenting a reference into its corresponding components (Figure 1). Given a reference, $\mathbf{x}$, as a sequence of $T$ tokens, $\mathbf{x} = \{x_1, x_2, ..., x_T\}$, CFE is the task of predicting the corresponding output sequence $\mathbf{y} = \{y_1, y_2, ..., y_T\}$ where each output symbol $y_i$ is one of $N$ possible bibliographic labels.

---

3. *Citation* can refer to both in-line citations or the entries in the references section of a paper. As *reference* is less ambiguous we will use this despite the name of the task.

4. URL withheld to preserve anonymity.

We experiment with both structured prediction models that use a linear-chain conditional random field [Lafferty et al., 2001] for prediction as well as an independent prediction model using RoBERTa-based representations. We summarized these models in this section.

**Structured Prediction Models**    The energy function for a particular configuration of the output sequence given the input is: $\mathcal{E}(\mathbf{y}|\mathbf{x}) = \sum_{t=1}^{T} \psi_{xy}(x_t, y_t) + \psi_{yy}(y_t, y_{t+1})$ where the emission score or the local log-potentials $\psi_{xy}$ is parameterized by a deep neural network, and the transition log-potentials $\psi_{yy}$ are parameterized by an input-independent parameter matrix. Modeling the intra-state dependencies under a Markovian assumption, we get the data log-likelihood as: $\log \mathbb{P}(\mathbf{y}|\mathbf{x}) = \mathcal{E}(\mathbf{y}|\mathbf{x}) - \log \sum_{\mathbf{y}'} \exp(\mathcal{E}(\mathbf{y}'|\mathbf{x}))$.

**Independent Predictions with RoBERTa**    Recent work on BERT and its variants [Devlin et al., 2019, Liu et al., 2019] has achieved state-of-the-art results on sequence tagging problems such as named entity recognition. We trained a RoBERTa model [Liu et al., 2019] to represent tokens in citation sequences. We then fine-tune this pretrained model for CFE task. For each token, the model makes an independent prediction of its class label. The prediction of the class label takes as input features of tokens from the last layer of the RoBERTa model and uses a linear classifier to predict the labels.

## 3. Automatically Generating CFE Data

BIBTEX allows researchers to specify all fields of the reference as a structured record and render the reference in a specified style. Figure 1 shows an example record in its key-value format. BIBTEX provides a natural way to automatically generate labeled training data for CFE models. The labels of each field are defined by the key-value record, which in turn give labels to rendered strings.

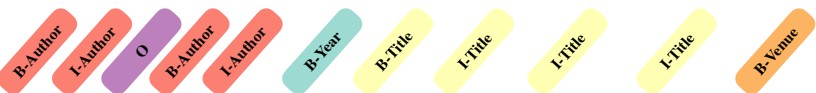

D. Kinga & M. Welling. 2014. Auto-encoding variational Bayes. ICLR.

```
@inproceedings{kingma:vae,
  title={Auto-encoding variational {Bayes}},
  author={Kingma, Diederik P and Welling, Max},
  booktitle={ Int. Conf. on Learning Representations },
  year={2014}
}
```

Diederik P Kingma and Max Welling. 2014. Auto-encoding variational Bayes. In *Int. Conf. on Learning Representations*.

Figure 1: An example BIBTEX entry and the corresponding citation generated with the natbib bibliographic style

### 3.1 Collecting BIBTEX

To create a large dataset of automatically labeled CFE examples, we collect BIBTEX records, curated by (and hosted on the websites of) researchers and organizations. Researchers tend to carefully curate their lists of publications due to the high value of these lists for job search and promotion. Collecting data from sources such as Google Scholar is problematic for several reasons, notably that manually curated records cannot be distinguished from automatically extracted ones.

We manually create a set of seed keywords from conference and journal listings websites, publishing venues and BIBTEX journal strings files. For example, some of our seeding words are "*Physics Letters*", "*IEEE Transactions on*", "*International Symposium on*" etc. Then, we construct a set of queries from these seeds and perform a web search to get BIBTEX files." We collected 6,023 BIBTEX files which contained 2,355,678 entries across various domains, a majority of entries being in Physics, Mathematics and Computer Science. Additionally, a smaller domain-specific dataset with more carefully chosen keywords was collected for analytical purposes.

### 3.2 Generating citation strings with labels

Different bibliography styles (`bst` files) can result in drastically different rendered strings. Figure 2 shows an example reference cited in different documents. To simulate this variation, we collect 269 BIBTEX styles from various sources such as CTAN and Overleaf. There are also a few options such as reference marker and hyphenation that can be modified from the TEX style rather than BIBTEX one. Due to the limitation of computing resources, we cannot render every reference for all styles. We run minimal set coverage on the citation fields defined by these styles and select 26 styles that cover all citation fields.

Figure 2: References using various BIBTEX styles

BIBTEX entries are randomly paired with BIBTEX styles and randomly chosen TEX options to produce citation strings in PDF format. As one can see from Figure 2, citation fields in the resulting string can be omitted, abbreviated or rearranged depending on the style being used. Therefore, aligning the citation field and its label is a non-trivial task. To address this problem, we generate an additional marked reference string (Figure 3). In the marked string, the citation field values are marked up with their corresponding labels. Then, we simply use these markers to detect segment labels and boundaries. Sometimes field values themselves change during the generation process, such as abbreviating author names. In this case, we use the string distance algorithms to complete the alignment.

<author>Krucker, S.</author> and <author>Benz, A. O.</author>, "<title>Heating events in the Quiet Solar Corona</title>," <journal>Proceeding of the Nobeyama Symposium<journal>, edited by <editor>T. S. Bastian</editor>, <editor>N. Gopalswamy</editor>, and <editor>K. Shibasaki</editor>, Vol. <volume>479</volume>, <month>December</month> <year>1999</year>, pp. <pages>25-30</pages>, <note>Provided by the SAO/NASA Astrophysics Data System</note>

Figure 3: An example marked reference string.

Finally, we extract citation strings from PDF files. In the PDF file, a citation string may spread across multiple lines and these line segments may jump around when being converted to text due to PDF extracting errors. To minimize noises causing by the PDF extraction process, we produce a single PDF for each citation. We run dataset generation in parallel and generate roughly 41M labeled citation strings. Table 1 shows the summary statistics for our dataset. In Table 2, we show the number of occurrences for some of the practical segment labels, both in the top high and low frequency. For the full segment counts of all 59 labels, see Appendix A. As shown in the tables, most of the labels of interest have a high number of supports.

| Parameter | BIBTEX dataset |
|---|---|
| Number of annotated references | 41,572,904 |
| Average reference length (in tokens) | 33.09 |
| Number of segment labels | 59 |
| Number of segments | 298,013,391 |
| Average segment length (in tokens) | 3.26 |
| Vocabulary size | 2,823,254 |
| Number of styles | 26 |
| Number of BIBTEX sources | 6023 |

Table 1: Summary of our BIBTEX CFE dataset.

| Label | Number of segments |
|---|---|
| author | 91,324,094 |
| year | 52,946,966 |
| title | 42,846,934 |
| journal | 20,620,003 |
| publisher | 9,777,982 |
| editor | 3,481,227 |
| institution | 1,928,709 |
| location | 3,125 |
| category | 219 |

Table 2: Segment counts for some labels of interest.

## 4. Experiments

**Model Details** We experiment with two model architectures: the standard **LSTM+CRF** for sequence labeling [Peters et al., 2018], and the **RoBERTa** architecture [Liu et al., 2019]. For brevity, we refer to the LSTM+CRF models by the features they use. We use a LSTM+CRF model with embedding features from **GloVe** [Pennington et al., 2014], **ELMo** [Peters et al., 2018] and **RoBERTa** [Liu et al., 2019] as word feature. We fix the word embedding features but train the LSTM and CRF parameters. The independent prediction model using RoBERTa is trained by fine-tuning all the weights of the RoBERTa model. We use the *fairseq* [Ott et al., 2019] implementation of RoBERTa.

Our hyperparameter settings for all LSTM+CRF models include a batch-size of 16 samples, bidirectional LSTM input size 896, hidden size 200 and a dropout rate 0.1 . We used Adam [Kingma and Ba, 2015] with learning rate 0.01 . For training the RoBERTa model with masked language modeling objective, we initialize the model with RoBERTa-base weights. We trained the model for 125 000 updates with Adam optimizer using polynomial decay learning rate scheduler, peak learning rate 0.0005 .

**Training datasets**    We compare: (1) **UMass CFE** and (2) UMass CFE+LM Pretraining+BIBTEX, denoted **+BIBTEX+LM**. UMass CFE refers to the human annotated data [Anzaroot and McCallum, 2013]. BIBTEX is our automatically labeled training data. We partitioned it into subsets according to their source bib entries. From these partitions, we sampled a 5M/45K training/ validation split instead of using all generated examples due to computation resource limitation. LM pretraining refers to training the RoBERTa model for token representations from all 41M citation reference strings using the LM objective [Liu et al., 2019]. The LSTM+CRF models are trained with UMass CFE data only. We compare training RoBERTa with these two schemes.

**Evaluation Datasets**    We evaluate on the **UMass CFE** dataset [Anzaroot and McCallum, 2013] has been the standard benchmark for CFE task for several years. It has 2476 references divided into 1454, 655, and 367 train/dev/test split. The examples from this dataset were extracted from ArXiv.org research papers prior to 2013 in PDF format and manually annotated. We also introduce two new automatically labeled collections of records for evaluation: **BIBTEX-Test** (held out references) and a collection of **domain-specific** subsets with a larger number of formatting styles.

**UMass CFE Results**    Table 3 shows span-level results of the various approaches. We find that the RoBERTa model trained on the BIBTEX data and UMass CFE data and with LM pretraining is the top performing system. It outperforms the state-of-the-art model [5] (Thai et al. [2018]) trained on UMass CFE data by **1.2** absolute point, on 17 out of 24 label classes and only perform worse on 3 classes. Table 4 compares individual label results.

| Model | UMass Dev | | | UMass Test | | |
|---|---|---|---|---|---|---|
| | P | R | F1 | P | R | F1 |
| Thai et al. [2018] | – | – | – | – | – | 0.951 |
| **GloVe** | **0.982** | 0.923 | 0.925 | 0.940 | 0.934 | 0.937 |
| **ELMo** | 0.954 | 0.947 | 0.950 | 0.955 | 0.946 | 0.951 |
| **BERT** | 0.941 | 0.932 | 0.936 | 0.932 | 0.925 | 0.928 |
| **RoBERTa** | 0.932 | 0.944 | 0.938 | 0.925 | 0.940 | 0.933 |
| **RoBERTa (+LM)** | 0.940 | 0.948 | 0.944 | 0.934 | 0.948 | 0.940 |
| **RoBERTa (+BIBTEX)** | 0.956 | 0.960 | 0.958 | 0.959 | 0.963 | 0.961 |
| **RoBERTa (+BIBTEX+LM)** | 0.954 | **0.964** | **0.959** | **0.960** | **0.967** | **0.963** |

Table 3: Span level results on UMass CFE dataset.

| | SoTa | Our | Δ |
|---|---|---|---|
| title | 0.9258 | **0.9661** | +0.0403 |
| publisher | 0.8525 | **0.9180** | +0.0655 |
| booktitle | 0.4416 | **0.6769** | +0.2353 |
| institution | 0.5455 | **0.9091** | +0.3636 |
| school | 0.5000 | **0.8000** | +0.3000 |
| year | **0.9944** | 0.9929 | -0.0015 |
| journal | **0.9583** | 0.9409 | -0.0174 |

Table 4: Per label F1 of **RoBERTa (+BIBTEX+LM)** compared to SoTA.

**BIBTEX-Test Benchmark**    Table 5 summarizes the performances of the best model on our BIBTEX CFE task on 600K held out references of the same domains and styles as the training corpora. The overall results is high (97%) due to the high accuracy of popular citation field labels. However, ambiguous labels (institution, school, organization) and some important labels with lower frequencies (edition, chapter), have room for improvement. We also evaluate our models on four domain-specific datasets. The results of our best model on these domain-specific dataset is shown in Table 6. We add data from additional styles, unseen at training time to each domain, which makes this dataset more challenging.

---

5. **ELMo** show comparable numbers for LSTM+CRF models.

| Labels | Precision | Recall | F1 | Count |
|---|---|---|---|---|
| author | 0.981 | 0.988 | 0.984 | 119,003 |
| title | 0.937 | 0.951 | 0.944 | 564,813 |
| year | 0.998 | 0.964 | 0.981 | 555955 |
| pages | 0.997 | 0.989 | 0.993 | 376960 |
| journal | 0.970 | 0.997 | 0.983 | 307,135 |
| volume | 0.994 | 0.986 | 0.990 | 232883 |
| institution | 0.889 | 0.832 | 0.860 | 22,558 |
| school | 0.893 | 0.873 | 0.883 | 12,271 |
| organization | 0.905 | 0.952 | 0.928 | 8,040 |
| edition | 0.876 | 0.551 | 0.677 | 1,538 |
| chapter | 0.960 | 0.582 | 0.725 | 1,278 |
| **overall** | **0.972** | **0.968** | **0.970** | **3,760,465** |

Table 5: Performance of **RoBERTa (+BIBTEX+LM)** a subset of citation field labels.

| Models | Math | | | Physics | | | Econs | | | CompSci | | |
|---|---|---|---|---|---|---|---|---|---|---|---|---|
| | P | R | F1 | P | R | F1 | P | R | F1 | P | R | F1 |
| **RoBERTa** | 0.832 | 0.809 | 0.820 | 0.860 | 0.803 | 0.831 | 0.832 | 0.784 | 0.807 | 0.858 | 0.810 | 0.833 |
| **RoBERTa (+LM-BIBTEX)** | **0.846** | **0.819** | **0.832** | **0.874** | **0.811** | **0.841** | **0.850** | **0.796** | **0.822** | **0.872** | **0.820** | **0.845** |

Table 6: Sequence tagger performances on selected domain.

**Generated citations quality.** To further assert the quality of the citation generation process, we manually checked the labeling sequences of 100 citations. We used the same annotation guidelines from [Anzaroot and McCallum, 2013]. We observed 96.607% Micro-F1 and 95.425% Macro-F1 with F1 scores on important fields such as *"title"*, *"author"* are 100% and 92.490%, respectively. The detailed report is shown in Appendix B. The lowest F1 score goes to the *"type"* field. We conjecture this to strings such as *"PhD thesis"* are often time not being tagged as *"type"* in the source BIBTEXentry.

## 5. Related work

The models for Citation Field Extraction largely rely on labeled data for training and evaluating. However, due to the ambiguous nature of the task, designing an annotation guideline and collecting the labeled dataset can be challenging. For example, "*school*" or "*institution*" can be used interchangeably. [Anzaroot and McCallum, 2013] released a hand-annotated citation field extraction dataset called UMass CFE which is hitherto the largest dataset for CFE, but it only contains ≈ 2000 annotated examples and limited unique segment labels. Our approach of reliably generating large amounts for

CFE with an exhaustive list of citation styles. Thus, our BIBTEX dataset provides a large amount of data for training deep learning models as well as an extensive benchmark for evaluating them.

Perhaps most closely related to our dataset generating mechanism, [Siegel et al., 2018] also use LaTeX as an auxiliary source to generate figures and their corresponding captions. While they employed several heuristics to locate captions and align them with figures, our method rely on the injected symbols to directly extract the segment labels.

Most previous works on CFE focus on modeling the global structure of the output space of label sequences. For example, [Anzaroot et al., 2014, Vilnis et al., 2015] learn hard and soft constraints on weights generated from templates. More recently, [Thai et al., 2018] employs a Latent Conditional Random Fields with learned embedding for output labels. While these models show improvements for learning in a limited data setting, we show that straight forward models trained on large, distant auto-generated data can achieve competitive performances.

## 6. Conclusion

We present an approach for automatically generating labeled data for citation field extraction using BIBTEX. We evaluate deep neural sequence labeling models trained on the data produced by our process and show improvements over models trained solely on standard human-annotated datasets [Anzaroot and McCallum, 2013]. The experimental results demonstrate the ability of the neural network models to generalize well given enough training data. We also evaluate performance on challenging automatically generated, domain-specific datasets, which are suitable as benchmarks in future work.

## Acknowledgements

We thank the anonymous reviewers for their constructive feedback. This material is based upon work supported in part by the Center for Data Science and the Center for Intelligent Information Retrieval, in part by the Chan Zuckerberg Initiative under the project Scientific Knowledge Base Construction, and in part using high performance computing equipment obtained under a grant from the Collaborative R&D Fund managed by the Massachusetts Technology Collaborative. Any opinions, findings and conclusions or recommendations expressed in this material are those of the authors and do not necessarily reflect those of the sponsor.

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

## A. BIBTEX dataset detail

| Label | Number of segments | Label | Number of segments |
|---|---|---|---|
| author | 91,324,094 | day | 10,109 |
| year | 52,946,966 | issue | 8,893 |
| title | 42,846,934 | archiveprex | 8,009 |
| pages | 30,002,992 | eid | 6,982 |
| journal | 20,620,003 | keyword | 4,726 |
| volume | 20,266,743 | primaryclass | 4,699 |
| booktitle | 15,283,924 | location | 3,125 |
| number | 10,195,511 | lccn | 2,622 |
| publisher | 9,777,982 | urldate | 1,290 |
| address | 8,031,345 | articleno | 867 |
| month | 6,940,739 | date | 764 |
| note | 3,813,340 | numpages | 671 |
| url | 3,552,654 | size | 516 |
| editor | 3,481,227 | annote | 395 |
| institution | 1,928,709 | collaboration | 279 |
| series | 1,367,418 | price | 255 |
| school | 1,012,928 | category | 219 |
| organization | 955,602 | paper | 209 |
| howpublished | 872,415 | city | 175 |
| type | 753,131 | advisor | 107 |
| doi | 356,624 | slaccitation | 93 |
| abstract | 293,142 | lastchecked | 85 |
| edition | 273,857 | intype | 54 |
| chapter | 236,182 | bookeditor | 28 |
| key | 215,691 | bookpages | 25 |
| issn | 145,376 | private | 24 |
| isbn | 137,646 | lastaccessed | 17 |
| eprint | 32,509 | translator | 5 |
| coden | 25,693 | version | 3 |
| comment | 12,105 | | |

Table 7: Segment counts of 59 labels.

## B. Quality of the Citation Generation Process

| Labels | Precision | Recall | F1 | Supports |
|---|---|---|---|---|
| author | 1.000 | 0.860 | 0.925 | 272 |
| year | 0.991 | 1.000 | 0.995 | 107 |
| title | 1.000 | 1.000 | 1.000 | 93 |
| publisher | 0.986 | 1.000 | 0.993 | 68 |
| journal | 1.000 | 0.957 | 0.978 | 23 |
| volume | 1.000 | 0.960 | 0.980 | 50 |
| pages | 1.000 | 1.000 | 1.000 | 74 |
| doi | 1.000 | 0.667 | 0.800 | 3 |
| school | 1.000 | 1.000 | 1.000 | 5 |
| address | 1.000 | 1.000 | 1.000 | 26 |
| booktitle | 2.000 | 1.000 | 1.333 | 30 |
| month | 2.000 | 1.000 | 1.333 | 12 |
| type | 1.000 | 0.250 | 0.400 | 4 |
| number | 1.000 | 0.947 | 0.973 | 19 |
| institution | 0.500 | 1.000 | 0.667 | 1 |
| url | 1.000 | 1.000 | 1.000 | 5 |
| editor | 1.000 | 0.905 | 0.950 | 21 |
| series | 1.000 | 0.800 | 0.889 | 5 |
| note | 0.778 | 1.000 | 0.875 | 7 |
| keyword | 1.000 | 1.000 | 1.000 | 1 |
| comment | 1.000 | 1.000 | 1.000 | 1 |
| chapter | 1.000 | 0.750 | 0.857 | 4 |
| organization | 1.000 | 1.000 | 1.000 | 2 |
| **overall** | **0.994** | **0.940** | **0.966** | **833** |

Table 8: F1-scores on 100 generated citations (manually checked).