# OpenReview forum: "Using BibTeX to Automatically Generate Labeled Data for Citation Field Extraction"
_AKBC.ws/2020/Conference — AKBC 2020_

### Official Review · AnonReviewer2 · 2020-03-27
**Nice contribution for AKBC - use distant supervision to label a large amount of data for CFE and improve performance**

**Rating:** 7
**Confidence:** 4

**Review:**

This paper is quite straightforward and makes a lot of sense for AKBC.
The authors note that the task of parsing reference strings is important nowadays for scientists and that the main dataset for training and evaluation is quite small.
The authors collect a large number of reference strings from multiple bib styles and label them using bibtex files. They they train a model on this large dataset and improve performance on the standard UCFE benchmark.

Overall this is very reasonable I have just two questions:

1. What is the accuracy of the automatic labeling procedure? Can some analysis be done?
2. The authors present various models that do not use their training data and one (roberta+bibtex+lm) that both uses their new bibtex data and also pre-trains with a MLM objective on their data prior to the downstream training. What is the contribution of the extra pre-training phase? If one just augments the data, it does not work well? This seems like an interesting and simple experiment to add.

---

> ### Author Response · Authors · 2020-04-17
> **Response to AnonReviewer2**
>
> Thank you for your time and careful review of the paper.
>
> 1. What is the accuracy of the automatic labeling procedure? Can some analysis be done?
>
> Q1 Accuracy of the automatic labeling procedure: Thank you for this suggestion, we will plan to add some analysis of this to a camera-ready version of the paper.
>
> 2. The authors present various models that do not use their training data and one (roberta+bibtex+lm) that both use their new bibtex data and also pre-trains with an MLM objective on their data prior to the downstream training. What is the contribution of the extra pre-training phase? If one just augments the data, it does not work well? This seems like an interesting and simple experiment to add.
>
> Q2: As we mentioned to AnonReviewer5, we have added a result comparing Roberta+LM Pretraining BibTex to Roberta+LM pretraining (without Bibtex). We found that Roberta+LM Pretraining+BibTex achieved better results. This highlights the importance and usefulness of our dataset. We will add an additional comparison to Roberta+Bibtex (without LM pretraining) to the camera ready, though we note that this is an ablation that does not test our main hypothesis of the usefulness of the BibTex labeled data.

---

### Official Review · AnonReviewer5 · 2020-03-30
**Acceptable results, but too much ambiguity in what they actually did**

**Rating:** 7
**Confidence:** 4

**Review:**

Update:  Authors have addressed my primary concerns around the value of the contributed dataset and the experiments.  I've revised my rating to Accept the paper

--------

The authors scraped about 6k BibTEX files from Math, Physics, and CS papers.  They use it to construct a larger, automatically-generated dataset alternative to the UMASS dataset for sequence tagging of bibliographic entries (e.g. title, authors, venue, year, etc.)  They perform some experiments and show some improvement on both UMASS and their new Bibtex-generated dataset.

My main critique of this paper is that there are critical details missing that are important to understanding what they did.  I’ve organized this below in 3 sections:
questions about the dataset that’s generated
questions about how they performed their experiments
questions about how they interpreted their results

----

Wrong/missing Citations:

In the discussion of data augmentation techniques, missing reference to Siegel et al “Extracting Scientific Figures with Distantly Supervised Neural Networks”.  Very related in that they use LaTeX files to generate large amounts of training data for neural models meant to process scientific documents.

Section 4 -- I think the reference to Peters et al 2018 for LSTM + CRF is probably wrong.  If you read Thai et al, the BiLSTM CRF baseline they’re using is from Lample et al 2016.


----

Questions about how they generated the dataset:

Figure 1 - how are you tagging periods, hyphens, or words like “In” that get added in front of the “booktitle” field?

“Then, we construct a set of queries from these seeds and perform a web search to get BIBTEX files.”  -- Can you be more specific how issuing web searches with resulted in BibTeX files?  What repositories/websites did you crawl?  How did you know which files were proper to grab?

The resulting dataset is 6k BibTeX files from Math, Physics and CS.  Are these not just mostly from arXiv LaTeX sources?  Could one simply have just downloaded the arXiv dumps and gotten way more BibTeX files?

Figure 3 -- Can you explain how the generation was done for the additional markup?  It’s quite an important distinction whether the LaTeX compiler is producing this output or you’re writing something custom to do this.

“To minimize noises causing by the PDF extraction process, we produce a single PDF for each citation” -- Nice idea

In Table 2 -- Might want to report the “year” field, since it’s 2nd most common field.

Because this is an automatically generated dataset, how do you assess the quality of this generated dataset?  One perspective is -- this is treated as data augmentation for improving training, in which case data quality doesn’t really matter as long as it helps the model perform on the UMASS test set.  But it also looks like you’re reporting performance on the Bibtex test set.  How does this dataset hold up as a test set in terms of quality vs human-curated UMASS?

------

Questions about how they performed their experiments:

In “Training datasets” section, did you mean “validation” instead of “cross validation” sets?  Seems unnecessary to perform cross validation on this size of dataset.

The UMASS dataset has 38 entity types:
6 coarse grain labels (reference_marker, authors, title, venue, date, reference_id).
“Venue” can be split into 24 fine grain labels (e.g. volume, pages, publisher, editor, etc)
Author and editor names can be also split into 4 finer-grain labels (e.g. first, middle, last, affix)
Dates can be segmented into year & month.

In your new BibTex dataset, there are 59 segment labels that don’t necessarily map to the labels in UMASS.  You’re missing discussion about how you consolidated these differences.  Did you map select labels in one dataset to another?  Or did you treat the two datasets as separate label spaces?  If so, how did you actually perform those Roberta experiments where you trained on both datasets?

Why no experiment without the LM pretraining?  Because you only performed UMASS vs (UMASS + LM Pretraining + Bibtex), it’s unclear any gains in the latter setting are due to continued LM pretraining to adapt Roberta to the bibliographic-entry-parsing domain, or due to the labeled data you’ve curated.

For example, if domain adaptation was key here (and the labeled data didn’t matter), could one have also performed LM pretraining on UMASS, or simply performed LM pretraining on a large number of bibliographic entries without going through all of the complicated Bibtex parsing to get labels?

The paper also lacks clarity about how the (UMASS + LM Pretraining + Bibtex) was performed.  Was there a particular order in which it happened?  Was it all done at the same time in a multitask manner?  How did you handle the vastly different sizes of the UMASS and Bibtex dataset?  How did you handle the differing label spaces?


------

Questions about how they interpreted their results:

It looks like LSTM+CRF with ELmo features is substantially better on UMASS than Roberta.  This makes me question whether the improvement from training on Bibtex could’ve also happened with an LSTM+CRF (elmo) model rather than Roberta?

Suppose the results showed that training on Bibtex helps for both Roberta & LSTM+CRF (elmo) models.  Then that would be a great result in favor of the Bibtex data being useful.  But if this result only held for Roberta, there might be something else happening here.

When reporting UMASS results, why does it say 24 classes?  UMASS has 38 classes.  What happened to the other classes?

Also, when referencing Table 4 results, why does it contain label types that aren’t in UMASS dataset (e.g. booktitle)?

What model is “SOTA” referring to in Table 4?   It’s not actually clear from Table 3.

---

> ### Author Response · Authors · 2020-04-17
> **Response to AnonReviewer5**
>
> Thank you for your time and careful review of our paper. We’ve responded to your questions in the order below. We hope that this provides additional clarity about our paper and the contributions of the paper.
>
> Thank you for your pointers to related work. These have been added to the paper.
>
> Dataset Generation -
>
> Q1: Regarding punctuation and works like “In” added to booktitle fields, we tag all additional words and punctuation added when BibTex is rendered as “O”.
>
> Q2: Regarding retrieval of BibTex from web searches, we use Google search to get personal curated bibliographies for each query term. The search is restricted to files of type “.bib”. For example, searching ‘NeurIPS filetype:bib’ gives us results such as http://www.cs.cornell.edu/people/tj/publications/joachims.bib
>
> Q3: Insufficiency of using arXiv latex source as a dataset. The bibtex files in the latex source of arXiv papers often contain bibtex records that directly copied from sources like Google Scholar and Semantic Scholar which themselves use automated citation extraction tools. These records could introduce noise and bias into our dataset. On the other hand, bibtex files curated by researchers, such as the bibtex files researchers keep for their research group’s papers are typically much more complete, accurate, and are manually curated. The distinctions of Math/Physics/CS stem from the seeding keywords used. We take conference/journal names and their abbreviations to form the seeding terms.
>
> Q4: Regarding the source of labels. We add special characters to the BibTex fields to denote starts and ends of spans. This changes the input to the LaTeX compiler and produces an output that easily provides the generated labels for the data. We don’t write custom rules for each style file (due to the relatively large number of styles) it might be a good idea to do so if there’s enough time and effort.
>
> Q5: Year is now reported in Table 2.
>
> Q6: Assessing the quality of the automatically labeled dataset. Thank you for this suggestion. We plan to add this a camera-ready version of the submission.
>
> Experiments -
>
> Q1 : Yes, we did mean validation, thank you for pointing this out. We have corrected it in the paper.
>
> Q2: Regarding the number of labels in UMass dataset, we used the experimental setup from Thai et al (2018). The original UMass dataset has 38 nested labels which is mapped to 24 labels as follows: we keep the coarse-grained labels for “author”, “title”, “date-year” and “reference-id”; we use the fine-grained labels under “venue” (2 of them overlap with the coarse-grained labels, 3 nested tag in “editor” as “editor” which is 20 labels in total), we also don’t use the “ref-marker” as it’s part of styling rather than a citation field (tagged as O).
>
> Q3: Regarding experiment on pre-training on BibTex and fine-tuning on UMass CFE, we use a single tag set in which we replace BiBtex labels that are not in UMass CFE with ‘O’ tags.
>
> Q4 & Q5. UMASS vs (UMASS + LM Pretraining + Bibtex) Experimental Setting -- We have added a result comparing Roberta+LM Pretraining BibTex to Roberta+LM pretraining (without Bibtex). We found that Roberta+LM Pretraining+BibTex achieved better results. This highlights the importance and usefulness of our dataset. We will add an additional comparison to Roberta+Bibtex (without LM pretraining) to the camera ready, though we note that this is an ablation that does not test our main hypothesis of the usefulness of the BibTex labeled data.
>
> Q6: Order of training objectives: First the language model is pretrained on the BibTex data. Then we fine-tune these pretrained representations for the citation field extraction task (combining both UMass and BibTex datasets together).
>
> Interpretation of results -
>
> Thank you for your time and careful review of the paper.
>
> Q1: Training LSTM+CRF with ELMo on BibTex dataset. We attempted to do this, but the LSTM+CRF model was too slow to train on the large dataset and so we were not able to report results.
>
> Q2: Bibtex training improvement for multiple models. This is a good point and would be nice to illustrate. We feel that our illustration for a single model demonstrates the importance and usefulness of this dataset, but would hope to extend this to multiple models in future work.
>
> Q3: Number of classes in the UMass dataset: Please see the answer to Q2 in Experiments above.
>
> Q4: ‘booktitle’ in the UMass dataset. We checked and confirmed that ‘booktitle’ is in UMass CFE dataset (unnested label of ‘venue-booktitle’).
>
> Q5: What model is “SOTA” referring to in Table 4?  It’s the ELMo model (comparable result with Thai et al. 2018).

---

> > ### Comment · AnonReviewer5 · 2020-04-22
> > **Questions have been address -- Will revise review to higher score**
> >
> > I thank the author for addressing the concerns I had above.  The two biggest concerns I had were: (i) Why this collected dataset is valuable (why not just grab everything from arXiv LaTeX bibs to get way more data?) and (ii) It's not obvious that the improvements are coming from training on the curated dataset VS simply more LM finetuning on any arbitrary Bibtex-looking text.
> >
> > Regarding (i), the authors response above discussing the pervasive reliance on Google/SemanticScholar generated BibTeX is quite interesting.  I don't necessarily buy the argument that this means scraping arXiv LaTeX bibs results in a worse quality dataset -- If everyone is using these tools to render their Bibliographies, then we should be training models to process Bib surface forms anyways; and that would be reflected in such a scraped dataset.  But still, it definitely makes the dataset they've collected more valuable & non-trivial to gather.
> >
> > Regarding (ii), the authors claim improvement on Roberta+LM+Bibtex over Roberta+LM (without Bibtex).  This ablation is basically what I was looking for, so I feel this concern has been addressed.  Regarding the authors note that Roberta+Bibtex is an ablation that doesn't really test the usefulness of the dataset, I suppose I don't necessarily share that same view.  It's unclear to me whether the value of such a dataset is in performing continued LM pretraining of Roberta onto this new text domain (Bibtex) or the actual task labels.  Such an ablation would help guide intuition.
> >
> > Anyways, regardless of my disagreements with some of the statements in the author response, the authors have indeed addressed my primary concerns, so I feel this paper should be accepted.

---

> > > ### Author Response · Authors · 2020-04-27
> > > **Thank you!**
> > >
> > > Thank you for your consideration of our response and for reconsider your reviewer score. We appreciate your time and effort reviewing and will add the Roberta+Bibtex (without LM pretraining)  ablation to the paper as well.

---

### Official Review · AnonReviewer4 · 2020-03-31
**A useful dataset and model for parsing bibliography entries.**

**Rating:** 6
**Confidence:** 4

**Review:**

The paper is releasing a dataset and a model for "citation field extraction", the task of parsing the a bibliography entry of a paper into its corresponding fields (authors, title, venue, year ... etc). The dataset is automatically generated by collecting 6k bibtex files and 26 bibtex styles, then compile pdfs for the bibtex entries using multiple different styles. The resulting dataset is 41M labeled examples. The model starts with the RoBERTa checkpoint, continue MLM pretraining on this data, then finetune as a sequence labeler, which results into new sota performance. Error analysis shows that there's a big difference between the model performance on common fields (authors, title, .. ) and less common ones (edition, chapter).

Pros:
- The task is important which makes the dataset a useful resource.
- Results are good compared to prior work

Cons:
- little novelty


Notes and questions:

- The dataset is constructed by randomly sampling a bibtex entry and a bibtex style, which means the dataset size can be easily increased or decreased. This brings a few thoughts:
1) how did you decide the dataset size of 41M examples?
2) Would a larger dataset make things better or a smaller dataset makes things worse? my guess is that the dataset is unnecessarily large, and you can get the same performance with a smaller one
3) Instead of random sampling of entries and styles, I will be curious to see if upsampling rare fields can improve their performance without loss in performance on other fields.

- Table5: the F1 numbers are not consistent with P and R.

- I will be curious to see how well grobid https://github.com/kermitt2/grobid does in this task, especially that it is, AFAIK, the leading PDF parsing tool.

- pretraining: how are the citations packaged into sequences of size 512? or do you train on individual examples?

---

> ### Author Response · Authors · 2020-04-17
> **Response to AnonReviewer4**
>
> Thank you for your thoughtful review of our paper and your time and consideration of our paper and this response. We have responded to your questions below:
>
> Q1: Dataset size - The size of the dataset was a constraint imposed by our computational resources. We had 1 TB of storage and created as large a dataset as possible with this storage. The bottleneck in storage was creating PDF files for each citation string.
>
> Q2: Dataset size vs performance - Recent trends in NLP have shown the benefits of pretraining models on massive corpora and so we decided to create and use as large a dataset as possible. That said, future work could investigate methods such as active learning to select a representative subset of data for more efficient training.
>
> Q3: Upsampling rare fields- Thank you. This is a very nice suggestion. We will be making code available for our work and hope that future work could explore this.
>
> Q4: Table 5 F1 values. - Thank you for pointing this out. We have fixed this and updated this in the paper.
>
> Q6: Comparison to Grobid - Thank you for this suggestion, we agree it would be interesting to make this comparison. We will look into what would be needed to make this comparison.
>
> Q7: Citation sequence length/batch size - We are able to fit most citation strings in 512 length sequences. We are able to train with a batch size of 6.
>
> Novelty concerns: Previous work on citation field extraction achieves state-of-the-art results using models that have complex dependencies in the output label space. In this work, we show that even better results can be achieved with simpler models that have been trained on much larger (automatically labeled) data. This has an important place in the recent trend of pretraining for NLP tasks. We achieve our results by designing a new method for automatically labeling data for citation field extraction.

---

### Decision · Program_Chairs · 2020-05-01

**Decision:**

Accept

**Comment:**

The authors consider the problem of citation field extraction, which is necessary for automatically constructing scholarly databases. While there were some concerns regarding the novelty of the work, the importance of the task and dataset to be released with the work outweights these.